# Prevalence and risk factors for *Trichomonas vaginalis* infection among adults in the U.S., 2013–2014

**Erin L. Tompkins**[1]*, **Thomas A. Beltran**[2], **Elizabeth J. Gelner**[3], **Aaron R. Farmer**[4]

**1** Department of Internal Medicine, Walter Reed National Military Medical Center, Bethesda, MD, United States of America, **2** Department of Clinical Investigation, Womack Army Medical Center, Fort Bragg, North Carolina, United States of America, **3** Department of Obstetrics and Gynecology, Bassett Army Community Hospital, Fort Wainright, Alaska, United States of America, **4** Department of Infectious Disease, Womack Army Medical Center, Fort Bragg Research Institute, Fort Bragg, North Carolina, United States of America

\* Erin.L.Tompkins.mil@mail.mil

**Data Availability Statement:** Data are available by the Centers for Disease Control and Prevention at https://wwwn.cdc.gov/nchs/nhanes/ContinuousNhanes/Default.aspx?BeginYear=2013

## Abstract

### Objective

*Trichomonas vaginalis* (TV) infection is common, curable, and associated with significant reproductive morbidity and risk for HIV infection. This analysis updates estimates of the prevalence of asymptomatic TV infection, and its associated risk factors, in the non-institutionalized U.S. population.

### Methods

We analyzed data from 4057 individuals who participated in the National Health and Nutrition Examination Survey (NHANES) 2013–2014 data collection cycle. Participant interviews ascertained demographic characteristics, self-reported tobacco use, and sexual history. Self-collected urine specimens from participants aged 18 to 59 years were tested for TV infection using the Gen-Probe Aptima TV assay. Cotinine was assayed from serum to provide a biomarker of recent tobacco exposure. Weighted percentages are provided to account for unequal selection probabilities among participants and adjustments for non-response.

### Results

Our sample included 1942 men (49.2%, 95% Confidence Interval [CI] 48.0–50.5) and 2115 women (50.8%, 95%CI 49.5–52.0). The infection prevalence among men was 0.5% (n = 16; 95%CI 0.2–1.0) and 1.8% (n = 55; 95%CI 1.1–3.1) in women. After controlling for participant characteristics associated with TV infection, females had a 5.2-fold increased odds of being infected compared to men (adjusted odds ratio (aOR) 5.2, 95% CI 2.4–11.4). Non-Hispanic blacks were more likely to be infected compared to non-Hispanic whites (aOR 11.2, 95% CI 4.6–27.2). Individuals below the federal poverty level were more likely to be infected compared to those earning >3 times the federal poverty level (aOR 6.7, 95% CI 1.7–26.6), and

**Funding:** The authors received no specific funding for this work.

**Competing interests:** The authors have declared that no competing interests exist.

active smokers were more likely to be infected compared to participants with no nicotine exposure (aOR 8.7, 95% CI 4.1–18.2).

## Conclusion

*Trichomonas vaginalis* infection continues to be relatively common, especially in women, smokers, non-Hispanic blacks, and in groups of lower socioeconomic status. Identifying the demographic characteristics of populations in the United States disproportionately affected by TV could impact screening and treatment of this infection in clinical practice. Further research on whether screening and treating for asymptomatic TV infection in high-risk populations improves risk for reproductive morbidity and HIV infection is warranted.

## Introduction

Sexually transmitted infections (STIs) are an important cause of morbidity in the United States. *Trichomonas vaginalis* (TV) is one of the top three non-viral STIs in the world. [1,2] Illness due to this pathogen can be overlooked by clinicians, as the disease process generally follows a benign course in both men and women, and is frequently asymptomatic. [3] Historically, studies focused on trichomoniasis in women, and some estimate that as many as 50% of TV infected women are asymptomatic. [4] While treatment is generally effective with a single dose of antibiotics, [5] many patients at risk for other STIs are not evaluated for TV, as the disease is not routinely included in STI testing, even in high-risk groups. [6]

In certain populations, TV is associated with significant disease burden, as well as adverse pregnancy outcomes. [7] Recently, less benign findings including infertility, chronic urogenital inflammation and increased transmission of HIV-1 have been associated with TV infection. [8,9] A prior review of National Health and Nutrition Examination Survey (NHANES) data from 2001–2004 estimated TV prevalence of 3.2% in adult females in the United States. [10] Prevalence is significant enough to suggest the screening of high-risk groups in clinical practice. [11] There are no current screening and surveillance guidelines for this disease in the general U.S. population, and some consider TV a neglected STI. [6] In this study, we discuss the TV prevalence data as well as examine demographic and behavioral characteristics associated with infection and suggest some clinical practice changes regarding screening for TV.

## Materials and methods

### Study design

This research was approved by the Walter Reed National Military Medical Canter Institutional Review Board (IRB# WRNMMC-EDO-2019-0354). Informed consent was not obtained because this dataset is deidentified and publicly available for analysis. We performed an analysis of data collected from the 2013–2014 NHANES cycle. The NHANES is a program administered by the Centers for Disease Control and Prevention (CDC) which is designed to assess the health and nutrition status of the non-institutionalized U.S. population. The program involves cross-sectional surveys of consenting participants, combining both household interviews and physical examinations. [12] Each two-year data collection cycle is a nationally representative sample designed to reflect the greater non-institutionalized U.S. population.

The NHANES sampling procedure is a complex multistage probability cluster design which oversamples specific populations such as Hispanics, non-Hispanic blacks, non-Hispanic

Asians, older adults, and low income persons to obtain both adequate samples for meaningful subgroup analyses as well as more reliable parameter estimates. [13] To account for unequal selection probabilities among participants and adjustments for non-response, all estimates were weighted using the provided sampling weights. The survey's design and weighting methodology have previously been described. [14] The data collection protocol was approved by the National Center for Health Statistics (NCHS) institutional review board. Per NCHS standards, all adult participants provided written informed consent. All information collected in the survey is kept strictly confidential and privacy is protected by law. [12]

## Demographic data and characteristics

Demographic information was obtained during the initial home interview. Pooling of categorical variables was conducted. Participant characteristics included gender, race/ethnicity (non-Hispanic white, non-Hispanic black, Hispanic, or other including multiracial), marital status (married or member of an unmarried couple; divorced, widowed, or separated; or never married), education level (did not graduate high school, graduated high school or attained a GED, or greater than high school education), health insurance coverage, and the participant poverty income ratio (PIR). The PIR was calculated by dividing family (or individual) income by the poverty threshold index adjusted for family size at the time of the interview. The poverty threshold index was provided by the Department of Health and Human Services and was specific to each survey year.

Behavioral risk factors were also examined for their relationship to TV infection. Sexual history was obtained from participants using an audio computer-assisted self-interview in the mobile examination center. Participants who reported having sex (defined as vaginal, oral, or anal sex) were asked follow on questions about their sexual history including the number of lifetime sexual partners and age at first sexual encounter. Additionally, participants were queried as to their smoking status (smoker or non-smoker).

## Specimen collection and laboratory methods

In the mobile examination center, participants aged 14 to 59 years were asked to provide self-collected urine specimens for analysis. After processing, portions of these urogenital samples were transported to the Division of AIDS, STD, and TB Laboratory Research, National Center for Infectious Diseases. There the samples were tested for TV infection using the Aptima TV assay (Gen-Probe Inc., San Diego, CA; [ATV] and analyzed using Panther system). A detailed description of the laboratory procedures and assay has been described previously. [15] At the time of this analysis, the ATV was the only FDA approved nucleic acid amplification test (NAAT) for TV. Data for FDA clearance established the ATV's sensitivity at 95.2% for urine samples with specificity for TV of ≥98.9%. [16] Although participants aged 14 to 17 provided samples for analysis, only results from adult samples (aged 18 to 59) were made publicly available and so included in this analysis. The Panther system is designed to test clinician-collected endocervical swabs, vaginal swabs, and specimens collected in solution. Analysis of urine from males and females may indicate off-label use for the Panther system.

Serum cotinine was used as a biomarker of recent tobacco use and environmental nicotine exposure. Cotinine is a primary nicotine metabolite widely used to assess nicotine exposure. This biomarker has been shown to better correlate with cigarette smoking than does self-report. [17] The elimination half-life of cotinine is 15 to 20 hours and thus is a useful measure of recent exposure. [18] Serum cotinine was measured by an isotope dilution-high-performance liquid chromatography/atmospheric pressure chemical ionization tandem mass spectrometry. [19]

Participants were categorized into three groups (smokers, environmental tobacco smoke [ETS] exposure, or non-smokers) using previously determined race and ethnicity based cut points for identifying adult smokers. Participants self-identifying as non-Hispanic white were classified as active smokers if their serum cotinine met or exceeded 4.85 ng/mL. Cut points for Non-Hispanic blacks, Hispanics, and all other races including multiracial, were 5.92 ng/mL, 0.84 ng/mL, and 3.08 ng/mL: respectively. Participants below these cut points but with serum cotinine levels 0.05 ng/mL or greater were classified having ETS exposure. Participants with serum cotinine levels below 0.05ng/mL were classified as non-smokers having no ETS exposure. Prior research estimates the sensitivity and specificity of these cut points to be greater than 95%. [20] Serum cotinine concentrations are reported as geometric means (ng/mL) with 95%CIs.

### Statistical analysis

To account for unequal selection probabilities among participants and adjustments for non-response, all estimates were weighted using sampling weights provided by the National Center for Health Statistics. Summary statistics are provided for categorical variables and include the number of participants as well as the weighted prevalence within each category. Weighted prevalence estimates are reported as percentages with 95% Wald confidence intervals (95%CI). Categorical variables were analyzed using Rao-Scott adjusted chi-square tests of independence.

Odds ratios (ORs) and adjusted odds ratios (aORs) with 95% CIs were estimated using multivariable logistic regression models. A backward elimination approach was used to produce a reduced model. Covariates in the reduced model included gender, race/ethnicity, education, PIR, and cotinine-based smoking status. *P*-values less than .05 were considered statistically significant. Data analyses were conducted using the complex sample package for SPSS 25 (IBM, Armonk, NY, USA).

## Results

Urogenital specimens from 4057 individuals were examined for TV infection. This sample included 1942 men (47.2%, 95%CI 48.0–50.5) and 2115 women (50.8%, 95%CI 49.5–52.0) representative of over 86 million men and 89 million women in the greater U.S. population. There was no significant difference in the proportion of men and women in the overall sample, *P* = 0.23. However, a significant difference was detected between the genders with regard to the overall infection prevalence, *P*<0.001. Among men, the TV infection rate was 0.5% (n = 16; 95%CI, 0.2–1.0). Among women the infection rate was 1.8% (n = 55; 95%CI, 1.1–3.1), almost 50% lower than what was reported among U.S. women in the 2001–04 cycle of NHANES. [10]

Table 1 shows TV prevalence by participant characteristics. The overall prevalence of TV in the sample was 1.2% (n = 71; 95%CI, 0.7–2.0). Infection with TV did not differ with respect to age, P = 0.91. The prevalence among non-Hispanic white participants was 0.4% (n = 10; 95% CI, 0.2–1.0); among Hispanic and Latino participants the prevalence was 0.3% (n = 4; 95%CI, 0.1–1.1), and among all other race/ethnicities, including multiracial participants the prevalence was 0.4% (n = 3; 95%CI, 0.2–0.8). Non-Hispanic black participants had the highest infection prevalence of the assessed race/ethnicity groups (n = 54; 6.8%, 95%CI, 4.0–11.2).

Individuals reporting a level of education less than a high school diploma or GED had a higher prevalence of infection (2.9%, 95%CI 1.5–5.6) compared to those who reported having graduated from high school or attained a GED (1.2%, 95%CI 0.6–2.2) as well as compared to those who reported education beyond high school (0.7%, 95%CI 0.5–1.2) (both P<0.05). TV

**Table 1. Prevalence of *T. vaginalis* infection by participant characteristics, n = 4057.**

| | | Total n | Weighted Prevalence % (95% CI) | P [a] |
|---|---|---|---|---|
| Gender | | | | |
| | Male [b] | 1942 | 0.5 (0.2–1.0) | <0.001 |
| | Female | 2115 | 1.8 (1.1–3.1) | |
| Age (yr) | | | | |
| | 18–25 [b] | 905 | 0.8 (0.4–1.6) | 0.49 |
| | 26–33 [b] | 724 | 1.5 (0.7–3.3) | |
| | 34–41 | 755 | 1.1 (0.7–1.8) | |
| | 42–49 | 776 | 1.2 (0.6–2.4) | |
| | 50–59 | 897 | 1.2 (0.6–2.6) | |
| Race c | | | | |
| | non-Hispanic White [b] | 1593 | 0.4 (0.2–1.0) | <0.001 |
| | non-Hispanic Black [b] | 822 | 6.8 (4.0–11.2) | |
| | Hispanic / Latino | 975 | 0.3 (0.1–1.1) | |
| | Other race—Including multiracial | 667 | 0.4 (0.2–0.8) | |
| Education c | | | | |
| | < High school | 831 | 2.9 (1.5–5.6) | <0.001 |
| | High school or GED [b] | 944 | 1.2 (0.6–2.2) | |
| | > High school | 2277 | 0.7 (0.5–1.2) | |
| Marital Status [c] | | | | |
| | Married or living with partner[b] | 2246 | 0.6 (0.3–1.1) | <0.001 |
| | Widowed, divorced, or separated | 553 | 2.0 (1.1–3.9) | |
| | Never married | 930 | 2.4 (1.3–4.3) | |
| Poverty to Income Ratio [c] | | | | |
| | Below federal poverty level | 964 | 3.9 (2.4–6.4) | <0.001 |
| | 1–3 times federal poverty level | 1397 | 1.1 (0.6–2.0) | |
| | >3 times federal poverty level | 1387 | 0.2 (0.0–0.6) | |
| Self-Reported Smoking status [c] | | | | |
| | Current smoker | 935 | 2.8 (1.9–4.1) | <0.01 |
| | non-smoker [b] | 635 | 0.7 (0.3–1.8) | |
| Age first had sex [c] | | | | |
| | <16 | 1026 | 1.8 (1.0–3.3) | <0.01 |
| | 16–18 | 940 | 0.9 (0.5–1.4) | |
| | >18 [b] | 912 | 0.4 (0.2–1.0) | |

[a]P value based on Rao-Scott adjusted chi-square statistic

[b]Estimate is unreliable due to relative standard error >30%

[c]Domain does not sum to 4057 due to participant non-response

infection also varied by marital status with participants who identified as married or living with partner having the lowest TV infection rate (0.6%, 95%CI 1.5–5.6). There was no difference in prevalence between individuals who identified as never married (2.4%, 95%CI 1.3–4.3) and those who identified as widowed, divorced, or separated (2.0%, 95%CI 1.1–3.9). However, both groups had a significantly higher prevalence compared to participants who identified as married or living with partner (both P<0.05).

No association was found between health insurance coverage and infection, *P* = 0.15; nor was there an association based on a participant's lifetime number of sexual partners, *P* = 0.11. In contrast, both income (as measured by the PIR) and a participant's self-reported age at first

sexual encounter showed associations with infection (both P<0.001). Regarding income, individuals reporting an income less than the federal poverty level had a prevalence of 3.9% (95% CI 2.4–6.4). The prevalence among those earning between 1 and 3 times the federal poverty level was 1.1% (95%CI 0.6–2.0) and among those reporting an income exceeding 3 times the federal poverty level the prevalence was 0.2% (95%CI 0.0–0.6). Individuals reporting their first sexual encounter prior to age 16 had a prevalence of 1.8% (95%CI 1.0–3.3) which was significantly higher than those who reported their first sexual encounter occurring between the ages of 16 and 18 (0.9%, 95%CI 0.5–1.4) or those who reported is occurring after age 18 (0.4%, 95% CI 0.2–1.0).

Smoking status assessed both through self-report and serum cotinine levels was found to be associated with TV infection. Self-described smokers were observed to be 3.4 (95%CI 2.1–5.6) times more likely to be infected with TV compared to self-described non-smokers (*P* <0.01). Examination of serum cotinine concentrations revealed a nearly ten-fold increase in TV infection among active smokers (OR 9.8, 95%CI 4.5–21.4) and nearly four-fold increase among individuals with ETS exposure (OR 3.9, 95%CI 1.3–11.8) as compared to participants with no significant nicotine exposure (P<0.001). Adjusting for gender, race/ethnicity, and differences in economic status as represented by the PIR, the aOR for active smokers dropped to 8.7 (95% CI 4.1–18.2) and the aOR for individuals with ETS exposure became non-significant (see Table 2). Thus, an individual's smoking status is an independent risk factor for TV infection. This data analysis, however, did not show a significant STI coinfection relationship. Among the 2174 participants who had tests results for both Chlamydia and TV, none were found to be co-infected with both.

Results of a multivariable analysis of TV infection risk factors are shown in Table 3. The following risk factors for infection were identified: gender (*P* < 0.001), race/ethnicity (*P* < 0.001), PIR (*P*<0.01), and smoking status (*P*<0.001). Pairwise interaction effects were observed for race/ethnicity and gender (*P*<0.001) as well as between race/ethnicity and PIR (*P*<0.001). Multivariable analyses indicate that after controlling for participant characteristics associated with TV infection, females had a 5.2-fold increased odds of being infected compared to men (adjusted odds ratio (aOR) 5.2, 95% CI 2.4–11.4). Additionally, non-Hispanic black participants were more likely to be infected compared to non-Hispanic white participants (aOR 11.2, 95% CI 4.6–27.2), individuals below the federal poverty level were more likely to be infected compared to those earning >3 times the federal poverty level (aOR 6.7, 95% CI 1.7–26.6), and active smokers were more likely to be infected compared to participants with no nicotine exposure (aOR 8.7, 95% CI 4.1–18.2).

## Discussion

*Trichomonas vaginalis* infection continues to be relatively common, especially in women, smokers, non-Hispanic blacks, and in groups of lower socioeconomic status. Based on this

**Table 2. Infection status by nicotine exposure.**

| Nicotine Exposure | TV Positive | | TV Negative | |
| --- | --- | --- | --- | --- |
| | n, % (95%CI) | Cotinine, ng/mL | n, % (95%CI) | Cotinine, ng/mL |
| Active Smoker | 43, 70.7 (61.1–78.7) | 284.39 (229.26–339.51) | 1166, 29.8 (26.3–33.6) | 218.05 (195.58–240.51) |
| ETS Exposure | 13, 16.6 (9.6–27.3) | 1.01 (0.25–1.77) | 760, 17.6 (15.3–20.3) | 0.56 (0.51–0.62) |
| No Exposure | 9, 12.7 (6.1–24.7) | 0.02 (0.01–0.03) | 1905, 52.1 (47.1–57.0) | 0.02 (0.02–0.02) |

a N = 3896

b Serum Cotinine concentration reflects the geometric mean

**Table 3. Unadjusted and adjusted odds ratios: *Trichomonas* infection risk factors, n = 3600.**

| Parameter | | OR (95% CI) | aOR (95% CI) [a] |
|---|---|---|---|
| Gender | | | |
| | Male | 1.0 (Referent) | 1.0 (Referent) |
| | Female | 3.8 (2.2–6.6) | 5.2 (2.4–11.4) |
| Race | | | |
| | non-Hispanic White | 1.0 (Referent) | 1.0 (Referent) |
| | non-Hispanic Black | 17.4 (6.5–47.0) | 10.3 (3.9–26.9) |
| | Hispanic / Latino | 0.8 (0.2–3.7) | 1.0 (0.2–4.6) |
| | Other race—Including multiracial | 1.0 (0.3–3.6) | 1.2 (0.4–4.1) |
| Poverty to Income Ratio | | | |
| | Below poverty level | 24.1 (7.5–77.4) | 6.7 (1.7–26.6) |
| | 1–3 times poverty level | 6.7 (1.4–32.9) | 2.9 (0.5–17.5) |
| | >3 times poverty level | 1.0 (Referent) | 1.0 (Referent) |
| Smoking Status [b] | | | |
| | Active Smoker | 9.8 (4.5–21.4) | 8.7 (4.1–18.2) |
| | ETS Exposure | 3.9 (1.3–11.8) | 2.3 (0.8–6.3) |
| | No ETS Exposure | 1.0 (Referent) | 1.0 (Referent) |

a Covariates include gender, race/ethnicity, PIR, and nicotine exposure

b Smoking status determined by serum cotinine concentration

analysis, in the United States, prevalence is approximated at 2.1 million people. By comparison, this exceeds U.S. chlamydia prevalence of 1.5 million people and gonorrhea prevalence of 0.5 million people. [21] Even when detected, this pathogen is not one of the nationally reportable diseases in the United States, making epidemiological study difficult. [22]

This analysis illustrates many demographic and clinical factors associated with a higher prevalence of TV, similar to prior studies. [23,24] Women, those with less than a high school education, those living below the Federal Poverty Line, and active smokers are significantly more likely to be infected with TV. Other less strongly associated risk factors included history of STIs, engaging in sex at less than 16 years of age, and single marital status.

Smokers in this data analysis were more likely to be infected with TV, as compared to non-smokers. We found that smoking status is an independent risk factor for TV infection when evaluating smoking status for this dataset through serum cotinine levels. The association between smoking status and STI infection has also previously shown to be true for Human Papilloma Virus, *Mycoplasma Genitalium*, and recently, in U.S. males with TV infection [24,25,26]. This may indicate a human biologic effect of smoking at the immune or local tissue level that impacts the propensity for infection.

The prevalence of TV infection did not differ by age category. This is in contrast to previous literature, and may be due to lower overall prevalence and sample size, possible shifting epidemiologic trends, changes in behavior patterns or due to difference in testing methods. [10] The urine samples included in this analysis were collected amongst largely asymptomatic individuals, the samples were selected randomly for TV testing and were not collected in the setting of a clinical exam. The 2013–2014 NHANES dataset was the first cycle to report data after initiating collection and analysis with the ATV test. While this method has proven sensitive for TV testing, it is worth noting that use of the ATV represents a difference in sample type, specimen collection and processing of samples as compared to previous years and may have had some yet unknown impact on study results. [12,15,26]

Prevalence of TV in this population did differ by gender; the prevalence in women is more than three times that of men. Higher prevalence in women has been consistently reflected in prior prevalence studies, although the cause or mechanism remains unclear. This analysis did not reveal a relationship between number of lifetime sex partners and infection, however, there was quite a low response rate for this measure. Only ten female participants reported number of lifetime sex partners and had valid TV test results in this dataset. Poverty is a risk factor in this analysis, but health insurance (or lack thereof) was not found to be a risk factor for infection with TV.

Identifying the demographic characteristics of populations in the United States disproportionately affected by TV could impact screening and treatment of this infection in clinical practice. [27] The associated disease burden of TV in women includes symptomatic vaginitis, urethritis, and can result in pelvic inflammatory disease. More concerning, infection with TV in women has been implicated as a cofactor in the pathogenesis of cervical cancer in some populations. [28] TV infection has been associated with infertility in both men and women with a postulated mechanism of inflammatory changes to the genital tract.

On a larger scale, TV infection poses a public health threat to pregnant women and neonatal health. [7] Although neither the American College of Obstetricians and Gynecologists, nor the United States Preventive Service Taskforce advocate for routine screening during pregnancy, TV has been associated with low birthweight infants and adverse pregnancy outcomes, including preterm labor as well as case reports of fatal neonatal brain abscess. [7,29]

Recent studies suggest that infection with TV may enhance acquisition and transmission of HIV-1 through increased inflammation and breakdown of the normal epithelial barrier. [8,9] Accurate diagnosis of this infection is also important as high levels of reinfection or persistent infection in high-risk groups has been documented. [5] International literature focused on women's health outcomes suggests that treatment and prevention of TV infection could reduce global HIV risk. [30] It is unclear how this would generalize to the HIV-positive population in the U.S. However, screening for TV infection in populations at risk for HIV may affect HIV transmission. [8,9]

Identification and treatment of TV infection could have an impact on U.S. morbidity and mortality, especially if targeting high-risk populations. Historically, TV has not been routinely tested as it is thought to be a minor, non-ulcerative, easily treatable infection. Given the high prevalence in select populations of U.S. women, clinicians may consider including TV in their screening algorithm for the prevention of transmission of other STIs, and reproductive morbidity (preterm delivery, infertility, pelvic inflammatory disease). [8] However, there is insufficient evidence to support such practice. There is one randomized trial that showed lack of preterm labor benefit of eradicating TV using metronidazole, resulting in a recommendation against screening and treating for TV in pregnant women. [31]

This study is limited by the self-reported nature of several of the key variables, such as the exclusion of minors, who are arguably a highly vulnerable population. Despite these shortcomings, NHANES data collection procedures are robust, and have been well validated for capturing accurate population sampling on which to ascertain prevalence estimates. Additionally, this is the first year for using NAAT results, rather than wet-mount sampling, and therefore should more accurately capture true rates of infection prevalence. This study suggests an opportunity to impact health outcomes with further research on the efficacy of screening and treating asymptomatic TV infection, especially in high-risk women.

The views expressed herein are those of the authors and do not reflect the official policy or position of the U.S. Army Medical Department, Department of the Army, Department of Defense, or the U.S. Government.

## Author Contributions

**Formal analysis:** Erin L. Tompkins, Thomas A. Beltran.

**Methodology:** Thomas A. Beltran.

**Project administration:** Erin L. Tompkins.

**Supervision:** Erin L. Tompkins.

**Writing – original draft:** Erin L. Tompkins, Thomas A. Beltran, Elizabeth J. Gelner, Aaron R. Farmer.

**Writing – review & editing:** Erin L. Tompkins, Elizabeth J. Gelner, Aaron R. Farmer.

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
