## [Decision Letter · Decision Letter 0]

18 Feb 2020

PONE-D-19-34688

Prevalence and Risk Factors for Asymptomatic Trichomonas vaginalis Infection Among Adults in the U.S., 2013-2014

PLOS ONE

Dear Dr. Tompkins,

Thank you for submitting your manuscript to PLOS ONE. After careful consideration, we feel that it has merit but does not fully meet PLOS ONE’s publication criteria as it currently stands. Therefore, we invite you to submit a revised version of the manuscript that addresses the points raised during the review process.

From my own reading of the manuscript, I agree with the reviewers. There are several items that need to be addressed but they seem surmountable and would greatly improve the manuscript.

We would appreciate receiving your revised manuscript by Apr 03 2020 11:59PM. To enhance the reproducibility of your results, we recommend that if applicable you deposit your laboratory protocols in protocols.io, where a protocol can be assigned its own identifier (DOI) such that it can be cited independently in the future. For instructions see: http://journals.plos.org/plosone/s/submission-guidelines#loc-laboratory-protocols

We look forward to receiving your revised manuscript.

Kind regards,

Ethan Morgan

Academic Editor

PLOS ONE

Journal Requirements:

Additional Editor Comments (if provided):

Per additional comments from one of the reviewers, please double check your associations: one of the major age associations (i.e., older individuals) that typically pops up did not pop up in this study.  Please also pay additional attention to the literature as both reviewers felt that more literature-based discussion of pertinent findings was necessary.

Reviewers' comments:

Reviewer's Responses to Questions

**Comments to the Author**

1. Is the manuscript technically sound, and do the data support the conclusions?

Reviewer #1: Yes

Reviewer #2: Yes

2. Has the statistical analysis been performed appropriately and rigorously? 

Reviewer #1: Yes

Reviewer #2: Yes

3. Have the authors made all data underlying the findings in their manuscript fully available?

Reviewer #1: Yes

Reviewer #2: Yes

4. Is the manuscript presented in an intelligible fashion and written in standard English?

Reviewer #1: Yes

Reviewer #2: Yes

5. Review Comments to the Author

Reviewer #1: Overall, nice report that will contribute to then literature on TV. Just a few very minor suggestions below, the primary concern at the moment is the introduction need (probably) just one more paragraph.

1. Review the intro for typos, there are just a few in there.

2. Even for me, the introduction may be a bit brief. Are there other studies that have examined demographic and behavioral characteristics? Could you summarize these some here? How is your study adding to past literature?

3. Table 1, please provide total N in title.

4. Table 3, footnote a seems unnecessary, but up to the authors.

5. Table 3, footnote b, is there a concentration threshold that should be specified here?

6. Results, paragraph 2, I would cite the 2001-04 NHANES cycle here.

Reviewer #2: In the manuscript “Prevalence and risk factors for asymptomatic Trichomonas vaginalis infection among adults in the U.S., 2013-2014” (PONE-D-19-34688), Tompkins et al. analyze a nationwide (asymptomatic) T. vaginalis prevalence assessment that was conducted under the auspices of the 2013-2014 National Health and Nutrition Examination Survey (NHANES). Noteworthy within this survey is the fact that nucleic acid amplification testing (in particular, transcription-mediated amplification) was used for laboratory diagnosis of T. vaginalis infection, rather than wet mount microscopy (as was used in past assessments). In this report, demographic associations with T. vaginalis detection included female gender, non-Hispanic blacks, individuals below the federal poverty level, and active smokers. No associations were made with participant age, as have been made in other published studies. Overall infection prevalence was reported at 1.2%, which the authors (in the Discussion section) extrapolated to 2.1 million individuals.

The manuscript was well written. Statistical methods appear appropriate. This report may potentially contribute valuable information toward national prevalence estimates of trichomoniasis. Specific comments follow:

1) The authors report an assessment of asymptomatic trichomoniasis. How did the authors determine that the patients were truly asymptomatic? Such documentation should be presented within the manuscript;

2) On which automated (Hologic) system were the APTIMA Trichomonas vaginalis Assays performed? As the authors likely know, male urine testing is FDA-cleared on TIGRIS instrumentation, while Panther automation (to the knowledge of this Reviewer) does not possess this indication. If performed with Panther automation, the authors may need to provide verification data, as such testing would be considered a laboratory-modified test. If performed on the TIGRIS, then the authors need to state as such.;

3) In line 158 and elsewhere, the authors note that their prevalence data reflect a 50% decrease over those reported in the 2001-2004 cycle. This Reviewer finds the comparison interesting, particularly in the context of the current data being generated from a molecular assay (one that has received indications for testing of both symptomatic and asymptomatic individuals). This finding (i.e., decrease) warrants additional discussion; and,

4) In lines 226-228, the authors provide an initial discussion of the association between T. vaginalis and smoking status. More discussion is warranted, supported by peer-reviewed literature references.

6. PLOS authors have the option to publish the peer review history of their article (what does this mean?). If published, this will include your full peer review and any attached files.

Reviewer #1: No

Reviewer #2: No

---

## [Author Response · Author response to Decision Letter 0]

6 Apr 2020

Reviewer #1: Overall, nice report that will contribute to then literature on TV. Just a few very minor suggestions below, the primary concern at the moment is the introduction need (probably) just one more paragraph.

1. Review the intro for typos, there are just a few in there. 

Thank you -- corrections have been made to most recent copy of the manuscript.

2. Even for me, the introduction may be a bit brief. Are there other studies that have examined demographic and behavioral characteristics? Could you summarize these some here? How is your study adding to past literature? 

I appreciate your comments -- I added additional background information and three resources (two of them newly published since this paper’s submission) which I feel added to the paper overall.

3. Table 1, please provide total N in title. This was added (n = 4057)

4. Table 3, footnote a seems unnecessary, but up to the authors. 

We left the footnote below the table to indicate to readers that the listed characteristics (all variables in the table) were included in the adjusted odds ratio (none were left out). 

5. Table 3, footnote b, is there a concentration threshold that should be specified here? 

It is important to specify serum cut points for cotinine concentration because they vary slightly by gender and ethnicity. These cut points are listed in the results section, lines 136-140. 

6. Results, paragraph 2, I would cite the 2001-04 NHANES cycle here. Thank you, this was added.

Reviewer #2: In the manuscript “Prevalence and risk factors for asymptomatic Trichomonas vaginalis infection among adults in the U.S., 2013-2014” (PONE-D-19-34688), Tompkins et al. analyze a nationwide (asymptomatic) T. vaginalis prevalence assessment that was conducted under the auspices of the 2013-2014 National Health and Nutrition Examination Survey (NHANES). Noteworthy within this survey is the fact that nucleic acid amplification testing (in particular, transcription-mediated amplification) was used for laboratory diagnosis of T. vaginalis infection, rather than wet mount microscopy (as was used in past assessments). In this report, demographic associations with T. vaginalis detection included female gender, non-Hispanic blacks, individuals below the federal poverty level, and active smokers. No associations were made with participant age, as have been made in other published studies. Overall infection prevalence was reported at 1.2%, which the authors (in the Discussion section) extrapolated to 2.1 million individuals.

The manuscript was well written. Statistical methods appear appropriate. This report may potentially contribute valuable information toward national prevalence estimates of trichomoniasis. Specific comments follow:

1) The authors report an assessment of asymptomatic trichomoniasis. How did the authors determine that the patients were truly asymptomatic? Such documentation should be presented within the manuscript; 

After reviewing the 2013-2014 CDC (NHANES) questionnaire again, it probably does not ask specific enough questions to ascertain whether or not participants are truly asymptomatic from an STI, as it does not ask questions specifically regarding dysuria, for example. Participants report existing conditions, but as it does not specifically ask about existing STIs, we omitted “asymptomatic” from the title of the manuscript.

2) On which automated (Hologic) system were the APTIMA Trichomonas vaginalis Assays performed? As the authors likely know, male urine testing is FDA-cleared on TIGRIS instrumentation, while Panther automation (to the knowledge of this Reviewer) does not possess this indication. If performed with Panther automation, the authors may need to provide verification data, as such testing would be considered a laboratory-modified test. If performed on the TIGRIS, then the authors need to state as such. 

The laboratory manual from CDC (https://wwwn.cdc.gov/nchs/data/nhanes/2013-2014/labmethods/TRICH_H_MET_TRICHOMONAS.pdf) states that the test was performed on the Panther system. The authors did not collect the data, and after review of the manual, it appears that the CDC performed quality control checks within their lab protocol. The NHANES dataset was downloaded from CDC and is open source, and was used for analysis in this manuscript. The lab analysis was performed by the Division of AIDS, STD, and TB Laboratory Research National Centers for Infectious Diseases National Centers for Disease Control and Prevention. Manual does not indicate that the test was modified from its intended use.

3) In line 158 and elsewhere, the authors note that their prevalence data reflect a 50% decrease over those reported in the 2001-2004 cycle. This Reviewer finds the comparison interesting, particularly in the context of the current data being generated from a molecular assay (one that has received indications for testing of both symptomatic and asymptomatic individuals). This finding (i.e., decrease) warrants additional discussion; and, We appreciate this comment, and have added additional context in the discussion portion of the manuscript. The lower prevalence in this sample may be due to sample size, or perhaps new/different testing methods used in analysis of this sample in comparison to years past. Higher prevalence among females was consistent in our calculations in comparison to existing literature and also consistent in previous years of NHANES datasets.

4) In lines 226-228, the authors provide an initial discussion of the association between T. vaginalis and smoking status. More discussion is warranted, supported by peer-reviewed literature references. 

Additional explanation of findings are discussed in lines 239-240, as well as three peer-reviewed articles on TV and other STIs are cited. We were also interested to find this correlation, and are excited to see that our calculated numbers from this dataset agree with the trend of smoking and TV infection found in other published works.

---

## [Decision Letter · Decision Letter 1]

17 Apr 2020

PONE-D-19-34688R1

Prevalence and Risk Factors for Trichomonas vaginalis Infection Among Adults in the U.S., 2013-2014

PLOS ONE

Dear Dr. Tompkins,

Thank you for submitting your manuscript to PLOS ONE. After careful consideration, we feel that it has merit but does not fully meet PLOS ONE’s publication criteria as it currently stands. Therefore, we invite you to submit a revised version of the manuscript that addresses the points raised during the review process.

Thank you for your initial revision. I agree with the reviewers comments below, particularly point number three by the second reviewer. I look forward to receiving these minor revisions shortly.

We would appreciate receiving your revised manuscript by Jun 01 2020 11:59PM. To enhance the reproducibility of your results, we recommend that if applicable you deposit your laboratory protocols in protocols.io, where a protocol can be assigned its own identifier (DOI) such that it can be cited independently in the future. For instructions see: http://journals.plos.org/plosone/s/submission-guidelines#loc-laboratory-protocols

We look forward to receiving your revised manuscript.

Kind regards,

Ethan Morgan

Academic Editor

PLOS ONE

Reviewers' comments:

Reviewer's Responses to Questions

**Comments to the Author**

1. If the authors have adequately addressed your comments raised in a previous round of review and you feel that this manuscript is now acceptable for publication, you may indicate that here to bypass the “Comments to the Author” section, enter your conflict of interest statement in the “Confidential to Editor” section, and submit your "Accept" recommendation.

Reviewer #1: All comments have been addressed

Reviewer #2: (No Response)

2. Is the manuscript technically sound, and do the data support the conclusions?

Reviewer #1: Yes

Reviewer #2: Yes

3. Has the statistical analysis been performed appropriately and rigorously? 

Reviewer #1: Yes

Reviewer #2: Yes

4. Have the authors made all data underlying the findings in their manuscript fully available?

Reviewer #1: Yes

Reviewer #2: Yes

5. Is the manuscript presented in an intelligible fashion and written in standard English?

Reviewer #1: Yes

Reviewer #2: Yes

6. Review Comments to the Author

Reviewer #1: All comments have been addressed, thank you for considering these revisions and responding to them in a timely manner.

Reviewer #2: In the manuscript “Prevalence and risk factors for Trichomonas vaginalis infection among adults in the U.S., 2013-2014” (PONE-D-19-34688R1), Tompkins et al. analyze a nationwide T. vaginalis prevalence assessment that was conducted under the auspices of the 2013-2014 National Health and Nutrition Examination Survey (NHANES). Noteworthy within this survey is the fact that nucleic acid amplification testing was used for laboratory diagnosis of T. vaginalis infection, rather than wet mount microscopy (as was used in past assessments). In this report, demographic associations with T. vaginalis detection included female gender, non-Hispanic blacks, individuals below the federal poverty level, and active smokers. No associations were made with participant age, as have been made in other published studies. Overall infection prevalence was reported at 1.2%, which the authors (in the Discussion section) extrapolated to 2.1 million individuals.

This Reviewer is examining this work for a second time. The authors addressed Reviewers’ concerns in appropriate fashion. The manuscript continues to be well written. Statistical methods were appropriate. This report may potentially contribute valuable information toward national prevalence estimates of trichomoniasis. This Reviewer has three final comments:

1) The authors revised the manuscript to indicate that testing of male urine was facilitated by the APTIMA Trichomonas vaginalis assay on Hologic Panther automation. The authors need to indicate in the text that this is off-label testing (i.e., a non-FDA-cleared specimen source on the Panther platform);

2) Line 121, the authors state that the APTIMA Trichomonas vaginalis assay is the only FDA-cleared nucleic acid amplification test for T. vaginalis. This may have been true at the time of the 2013-2014 NHANES assessment, but it is no longer true today. Please revise; and,

3) More explanation needs to be considered regarding the ~50% reduction in T. vaginalis prevalence. With conversion from a poorly-sensitive wet mount detection method to a highly-sensitive molecular assay (with performance characteristics well described in the peer-reviewed literature) among approximately 4000 participants (utilizing a well-defined means of participant selection), the quite significant reduction in prevalence is likely not due to the testing modality. Are changes related to epidemiology? Are changes related to treatment…due to the fact that perhaps more individuals are being tested by more-accurate molecular assays in a clinical setting? Please discuss.

7. PLOS authors have the option to publish the peer review history of their article (what does this mean?). If published, this will include your full peer review and any attached files.

Reviewer #1: No

Reviewer #2: No

---

## [Author Response · Author response to Decision Letter 1]

1 Jun 2020

Thank you for your comments and thorough review -- the following edits have been made (track changes comments below):

Reviewer #2: This Reviewer is examining this work for a second time. The authors addressed Reviewers’ concerns in appropriate fashion. The manuscript continues to be well written. Statistical methods were appropriate. This report may potentially contribute valuable information toward national prevalence estimates of trichomoniasis. This Reviewer has three final comments:

1) The authors revised the manuscript to indicate that testing of male urine was facilitated by the APTIMA Trichomonas vaginalis assay on Hologic Panther automation. The authors need to indicate in the text that this is off-label testing (i.e., a non-FDA-cleared specimen source on the Panther platform); 

Reply: comments regarding the source and testing methods were added to the methods section in lines 126-128.

2) Line 121, the authors state that the APTIMA Trichomonas vaginalis assay is the only FDA-cleared nucleic acid amplification test for T. vaginalis. This may have been true at the time of the 2013-2014 NHANES assessment, but it is no longer true today. Please revise; and, 

Reply: this was corrected in lines 121-122.

3) More explanation needs to be considered regarding the ~50% reduction in T. vaginalis prevalence. With conversion from a poorly-sensitive wet mount detection method to a highly-sensitive molecular assay (with performance characteristics well described in the peer-reviewed literature) among approximately 4000 participants (utilizing a well-defined means of participant selection), the quite significant reduction in prevalence is likely not due to the testing modality. Are changes related to epidemiology? Are changes related to treatment…due to the fact that perhaps more individuals are being tested by more-accurate molecular assays in a clinical setting? Please discuss. 

Reply: Additional comments were added to the discussion section, please see lines 245-249.

---

## [Editor Report · Decision Letter 2]

2 Jun 2020

Prevalence and Risk Factors for Trichomonas vaginalis Infection Among Adults in the U.S., 2013-2014

PONE-D-19-34688R2

Dear Dr. Tompkins,

We’re pleased to inform you that your manuscript has been judged scientifically suitable for publication and will be formally accepted for publication once it meets all outstanding technical requirements.

Kind regards,

Ethan Morgan

Academic Editor

PLOS ONE
---

## [Editor Report · Acceptance letter]

5 Jun 2020

PONE-D-19-34688R2 

Prevalence and Risk Factors for *Trichomonas vaginalis* Infection Among Adults in the U.S., 2013-2014 

Dear Dr. Tompkins:

I'm pleased to inform you that your manuscript has been deemed suitable for publication in PLOS ONE. Congratulations! Your manuscript is now with our production department. 

Kind regards, 

on behalf of

Dr. Ethan Morgan 

Academic Editor

PLOS ONE